# Traditional and New Microorganisms in Lactic Acid Fermentation of Food

Barbara Sionek [1] , Aleksandra Szydłowska [1],* , Kübra Küçükgöz [1] and Danuta Kołożyn-Krajewska [1,2],*

1 Department of Food Gastronomy and Food Hygiene, Institute of Human Nutrition Sciences, Warsaw University of Life Sciences (WULS), Nowoursynowska St.159C, 02-776 Warsaw, Poland; barbara_sionek@sggw.edu.pl (B.S.); kubra_kucukgoz@sggw.edu.pl (K.K.)

2 Department of Dietetics and Food Studies, Faculty of Science and Technology, Jan Dlugosz University in Czestochowa, Al. Armii Krajowej 13/15, 42-200 Częstochowa, Poland

* Correspondence: aleksandra_szydlowska@sggw.edu.pl (A.S.); danuta_kolozyn-krajewska@sggw.edu.pl (D.K.-K.)

**Abstract:** Lactic acid fermentation is one of the oldest and most commonly used methods of bioconservation. This process is widely used for food preservation and also for a production technique that relies on the metabolism of lactic acid bacteria (LAB) to convert carbohydrates into lactic acid. This fermentation imparts unique flavors and texture of foods, extends their shelf life, and can offer health benefits. There are both traditional and new microorganisms involved in the lactic acid fermentation of food. The current review outlines the issues of fermented foods. Based on traditional fermentation methods, a broad panorama of various food products is presented, with the microorganisms involved. The methods of both traditional fermentation (spontaneous and back-slopping) as well as the importance and application of starter cultures in mass food production are presented. Currently, based on the results of scientific research, the health-promoting effect of fermented foods is becoming more and more important. This is due to the presence of probiotic microorganisms that are naturally presented or may be added to them, as starter cultures or additives, and from the presence of prebiotics and postbiotics. New innovative methods of using probiotic microorganisms open up new and broad perspectives for fermented functional foods.

**Keywords:** lactic acid bacteria; starter cultures; probiotics; fermented food products

## 1. Introduction

Fermentation is one of the oldest processes in food technology. It is natural, ecological and leads to products with a wider taste range and higher nutritional value. This is a process confirmed by the history of thousands of years of successful food preservation. It meets the preferences and expectations of consumers and continuously gains more and more attention and popularity [1]. Lactic acid fermentation bacteria cause rapid acidification of the raw material through the production of beneficial organic acids, mainly lactic acid, but also acetic acid, ethanol, aromas, enzymes, and bacteriocins [2].

In the beginning, spontaneous fermentation was carried out with the participation of microorganisms naturally found in raw materials. In this case, however, the composition of the microflora participating in fermentation processes was not known. Spontaneous fermentation is still used, mainly in the production of fermented vegetables, fruits, fish, and traditional, raw-ripened meat products. An innovative solution was the use of so-called "starter cultures", which limited the development of the dangerous phenomenon of the development of wild bacterial strains [2]. Starter bacteria are used in the production of most fermented products: dairy, meat, cereals, and oriental products.

In recent years, new microorganisms and probiotic bacteria have been used in food production, thus obtaining a new type of products called functional food. Probiotic bacteria can be introduced as starter cultures or in addition to traditional bacteria. That way,

products with completely new properties are obtained, containing live colonies of intestinal bacteria, which, through the ability to colonize the digestive tract, influence various metabolic processes of the body and improve human health. It has also been demonstrated that probiotics may have certain advantageous qualities because of compounds they secrete, byproducts of their metabolism, or molecules released when their cells lyse. These substances are referred to as paraprobiotics, non-viable probiotics, inactivated probiotics, ghost probiotics, or bacterial metabolites, postbiotics, metabiotics, cell-free supernatants, or metabolic residues of probiotic action [3].

In the case of food with the addition of probiotic bacteria, the beneficial effect on the human body depends not only on the strain of bacteria but also on the medium used to introduce them. A food matrix is a raw material into which probiotic bacteria can be introduced and a food product can be obtained. The introduced probiotic bacteria may cause fermentation or will be present in a non-fermented product. In both cases, the challenge is to keep as many probiotic bacteria alive as possible.

One of the most commonly used carriers is dairy products. Other drinks with the addition of probiotic bacteria have also gained great popularity. These include, among others, fruit and vegetable juices. A relatively new direction is the introduction of probiotic bacteria into meat products. Fermented products with probiotic bacteria differ sensorially from those produced in a traditional way.

In newer products, the additives are the so-called prebiotic substances, e.g., carbohydrates, such as lactulose, fructooligosaccharides, and galactooligosaccharides, which selectively stimulate the growth of probiotic bacteria in the gastrointestinal tract [4].

## 2. The Historical Overview of Lactic Acid Fermentation in Food Production

Fermentation is one of the most effective traditional methods of food preservation, which for millennia has provided people around the world with rational management of valuable food resources. Perishable food and beverage products are transformed through fermentation into stable and safe products that can be stored and transported. In addition, fermented foods gain a number of organoleptic qualities and are more easily digestible. Fermentation prevents food spoilage, meaning increased availability and reduced food waste. Since the dawn of time, humans have used microorganisms to preserve food—from ancient civilizations such as Assyria, Babylonia, Egypt, and Persia to Slavic ones. Over the centuries, experience has been gained in which products can be ensiled, which speeds up the whole process, and what causes their spoilage—as well as learning methods that extend the durability of food products [5].

Due to the long history of effective usage of food fermentation, beginning in ancient civilizations, it should be acknowledged that fermentation is a great human cultural heritage. Fermentation is recognized as a safe method of food preservation and is widely accepted in the household and artisanal production of indigenous food products. Native fermentation-related microorganisms are naturally found in foods. In accordance with tradition, they can also be added in the form of batches of fermented food, the so-called back-slopping technique. Nowadays, at the industrial level, selected starter cultures are preferred to achieve the reproducibility of the final product. Tamang (2010) assessed that, nowadays, more than 5.000 various fermented food products are consumed worldwide [6].

The earliest archeological evidence of fermented beverage consumption came from Israeli stone mortars dated as being from 13,000 BC [7]. Other archaeological proof came from investigations in Qiaotou, China. It is suggested that 9.000–8.700 cal. BP fermented beer was made from rice and was consumed during prehistoric funerary rituals [8]. In human history, the fermentation of milk was a very important step, which enabled humans to obtain non-perishable food. Dairy products are a valuable source of protein. Archaeological evidence indicates that during the early Neolithic (6500 BC), in the Near East, milk was processed by the people. In Poland, evidence of dairy residues was found in pottery "cheese-strainers" dated around 5.200 and 4.900–4.800 BC (European Linear Pottery culture), which suggests milk fermentation for cheese making [9]. In ancient Egypt, a

variety of fermented foods, including bread, wine, and beer, were part of the diet. For centuries, hunting and fishing were primary sources of food [10]. The preservation of meat excess was of great importance to help people survive periods of food shortage. Fermentation, along with curing, smoking, salting, and sun-drying, is one of the conservation techniques transforming meat into a longer-lasting product. To save meat for the winter, Romans used the methods of meat preservation of the Gauls and Celts [11]. They further developed them, laying the foundations for a whole range of fermented Mediterranean meat products. Food fermentation and preservation run along with the development of human civilization. An example is lactic acid-fermented cabbage consumed around the ancient world: in the Roman Empire, as sauerkraut; in Asia, as a Chinese PaoCai gundrunk; in the Himalayas or in Korea, as a kimchi [12–14]. The history of kimchi is around 3000 years. Nowadays, it is an emblematic health-promoting product of the Korean diet and, since 2013, has been acknowledged as a Korean heritage culture product [15]. In the 18th century, Commodore John Byron and Captain James Cook, among others, found that rich-in-vitamin-C sauerkraut is an effective antiscorbutic agent of great value for lengthy sea voyages [16].

## 3. The Traditional Application of Lactic Acid Fermentation in Food Production

### 3.1. Dairy Products

Bacteria are one of the oldest living microorganisms on Earth. Despite their simple structure compared to eukaryotic organisms, they play an important role in nature, participating in the circulation of elements or the distribution of dead matter. These organisms have also been used in food production. The main food-fermenting microorganisms, irrespective of food kind, are LAB species. The data from the whole genome sequence distinguish 261 LAB species grouped in 26 genera (http://lactotax.embl.de/wuyts/lactotax/) accessed on 10 October 2023 [17]. Lactic acid fermentation bacteria are most commonly used in the food industry. The food industry uses LAB to produce food and probiotic cultures to improve food quality. Some LAB species are probiotic strains with health advantages.

LAB has a long history of applications in milk processing and preservation. Natural fermentation of milk with LAB converts lactose into lactic acid, leading to the acidification of dairy products. Lactic acid production during fermentation is essential because it results in a pH decrease, creating an environment in which pathogens cannot exist, and the spoilage is inhibited [18]. LAB degrades milk proteins into peptides and amino acids, and transforms lipids into free fatty acids. Fermentation influences the physical and chemical properties of dairy fermented products and is responsible for their specific texture and rheological parameters and contributes to flavor, shelf life, and safety [19].

The traditional, natural milk fermentation methods include spontaneous, native LAB fermentation or the techniques of back-slopping, where the fermented product is added to the new batch to facilitate the next process. In the back-slopping technique, a successful batch is reused, contributing to the selection of the best starter cultures for the quality of fermented products [20]. However, the implementation of empirical and artisanal techniques often leads to inconsistent food products [21]. In traditional dairy products, raw milk is used for containing a variety of autochthonous microorganisms, including different LAB strains. The natural and safe preservatives of traditional dairy products are ensured by LAB-produced antimicrobial peptides called bacteriocins [19]. According to Limosowtin et al. (1996), a starter culture can be divided into "Defined Strain Starters" including commercial single strain starters and "Mixed Strain Starters" including, among other categories, "Artisanal or Natural Mixed Starter Cultures" [22]. Microflora of mixed starters contain a wide range of microorganisms; however, LAB species are predominant [23]. Nowadays, the above-mentioned techniques are replaced in the dairy industry via the addition of specific starter cultures for inoculation, which gives the possibility to ensure the safety and predictability of fermented products. Starter cultures are selected by considering features, such as hetero- and homofermentation, probiotic properties, acid tolerance, organic acid production, temperature range, salt tolerance, oleuropein separation ability, flavor devel-

opment, and bacteriocin production, in order to dominate the natural microbiota [24]. In the case of the use of traditional starter cultures together with intestinal bacteria, the new fermented products obtained as a result of fermentation are classified as second-generation beverages. When intestinal strains are exclusively used, the new fermented product is recognized as a third-generation beverage [25].

The dairy fermentation starter cultures contain mainly lactic acid bacteria; other microorganisms like yeasts and molds are rarely used. The following hygienic standards for pasteurized milk are used on an industrial scale. The development of molecular biology allowed for the detection, typing, and identification of microorganisms, and, consecutively, the commercially produced starters consist of well-selected high counts of suitable strains [26]. Recently, there has been a growing interest in exopolysaccharides producing LAB and their applications in the food industry. Exopolysaccharides influence interactions with milk proteins and micelles, resulting in improvements in casein network firmness and ability to bind water [27]. It contributes to unique rheological and sensory characteristics of dairy-fermented products.

Many distinct products can be obtained as a result of milk fermentation with huge diversity in organoleptic traits according to the fermented microorganisms, recipes, and additives. The main dairy-fermented products are yoghurts, kefirs, and cheeses. Yoghurt is a commonly consumed dairy product and is made by fermenting milk. Mostly *Lactobacillus bulgaricus* and *Streptococcus thermophilous*, *Lactobacillus acidophilus* and *Bifidobacterium bifidum* are isolated from yoghurt samples. The fermentation process not only causes a sour taste but also increases the health benefits, such as promoting gut health and digestion [28]. In yoghurt and yoghurt-like products, moderate thermophile LAB as starter cultures are produced: *Lactobacillus delbrueckii subsp. bulgaricus* and *Streptococcus thermophilus* [29]. These strains also produce exopolysaccharides, which have an impact on reduced syneresis, increased viscosity, smoother, and creamier texture of the yoghurt. Kefir is a fermented beverage made from milk or water using kefir grains, consumed in different Asian countries. "Kefir name is produced from the Turkish word "keyif" and means "good feeling" [30]. Kefir grains contain a diverse community of bacteria and yeast, including *Lactobacillus kefiranofaciens*, *Lacticaseibacillus paracasei*, *Lactiplantibacillus plantarum* and *Saccharomyces cerevisiae*. This diversity contributes to the complexity of kefir's flavor and probiotic content [31]. During the fermentation process of kefir, lactose is converted by lactic acid bacteria into lactic acid, which gives kefir its characteristic flavor. The yeasts in kefir grains also contribute to the fermentation process by producing ethanol and carbon dioxide, giving kefir its effervescent structure [32]. The fermentation of kefir not only enhances its flavor but also leads to the formation of various bioactive compounds, such as vitamins, organic acids, and peptides, which contribute to its nutraceutical benefits [33].

Dairy beverages are very popular fermented milk products, widely consumed in the whole world and, nowadays, they are recognized as vehicles for probiotic delivery [34]. It is essential to select the right probiotic strains of viability and functional activity throughout the shelf life of the product. The success of probiotic milk beverages can be limited by the nature of the ingredients and the fear of contamination as well as low strain viability during storage.

The viability of bacteria of the former genus *Lactobacillus* spp. and *Bifidobacterium* spp. in milk beverages, including the probiotic strains used, can be influenced by many factors: pH, presence of hydrogen peroxide and dissolved oxygen, concentration of metabolites such as lactic and acetic acids, buffering capacity of the medium, storage temperature, and type of added ingredients [35].

There are at least 1000 varieties of cheeses available, which can be classified into the following: soft-to-hard bacterially ripened internally, bacterial surface-ripened, and mold-ripened [36]. Due to exceptional sensory properties, regional, low-processed cheeses are becoming more popular [37]. The fermentation of dairy products can be conducted by an unknown mix of adventitious or environmental microbiota or via the addition of microbial inoculation. Spontaneous fermentation can be optimized by using the back-slopping

technique, which provides a higher number of suitable and best-adapted-for-fermentation microorganisms. The advantages of back-slopping include maintaining microbes' genetic biodiversity. This technique is still widely used at home and in artisanal whey and cheese manufacturing [26]. The traditional manufacturing process of raw milk fermentation with diverse microbial compositions has a direct impact on the final product, maintaining its uniqueness. In regional cheesemaking, the predominance of LAB and hygienic standards are key determinants in reductions in undesirable pathogenic and spoilage microorganism occurrence, resulting in product safety [38]. During further manufacturing (ripening) and maturation of non-LAB, secondary flora can occur, influencing the original cheese [39].

In large-scale cheesemaking, the usage of starter cultures ensures control over fermentation and contributes to product standardization and safety. However, the use of commercially selected starter cultures strictly limits biodiversity and influences sensory properties of produced cheeses. The direct addition of a large number of cells of selected microorganisms rapidly and successfully acidifies pasteurized milk and inhibits other microorganism occurrence, including secondary microflora. Moreover, the wide use of commercially selected starter cultures may lead to the disappearance of certain wild strains, leading to the loss of genetic material [40]. In the cheesemaking industry, starter bacteria according to the produced cheese can be divided into groups: mesophilic like *Lc. lactis subsp. cremoris* and *Lb. lactis* ssp. (Cheddar, Gouda, Edam, Blue, and Camembert) or thermophilic, including *Lb. delbrueckii* ssp. *delbrueckii*, *Lb. delbrueckii* ssp. *bulgaricus*, *Lb. delbrueckii* ssp. *lactis* or *Lb. helveticus* (Emmental, Gruyere, Parmesan and Grana). The most-often used starter bacteria are members of the genera *Lactococcus*, *Lactobacillus*, *Streptococcus*, *Leuconostoc*, and *Enter-coccus* [39]. In cheesemaking, starters contribute significantly to the beginning of manufacturing, in the young curd. In the further process of cheese production, which takes many months or even years of ripening, the microflora changes. Subsequently, secondary microflora appears and dominating microorganisms are non-starter LAB. The population of non-starter LAB consists of mesophilic and thermophilic lactic acid bacteria and *lactococci*, *pediococci*, *enterococci*, and *Leuconostoc* spp. [41]. The composition of many cheese types' microbiota differs; therefore, the non-starter LAB significantly influences the cheese quality traits unique for regions and factories.

In accordance with scientific evidence, fermented dairy products are one of the best available health-beneficial natural products, which are well tolerated without causing undesired side effects [42]. Fermented dairy products have immunomodulatory and antioxidant activity. The results of numerous studies and clinical trials suggest their potential health advantages, including anticancer, antihypertensive, and anti-cholesterol effects.

### 3.2. Fermented Plant-Origin Foods

Fresh fruits and vegetables are prone to spoilage and have short shelf life. Therefore, preservation is a very important issue in human nutrition. Fermented foods are a common source of many essential nutrients available to the population in many countries around the world. Fermented fruits, vegetables, and cereals are excellent examples of native food products with high nutritional value. They are primary and rich sources of fiber, carbohydrates, vitamins (thiamine, riboflavin, niacin, or folic acid), and minerals (iron, calcium, and phosphorus), and other valuable nutraceuticals, such as polyphenols, tannins, flavonoids, and flavonols [43]. Due to seasonal availability of fruits and vegetables, fermentation allows for the preservation of this valuable food during winter months. Moreover, fermentation processes increase the nutritional value and improve the bioavailability of minerals and vitamins, as well as contributing to food safety [44]. Fermentation of plant-origin food products is an alternative to the methods and techniques including chemicals used in the food industry.

Fruits and vegetables have a vast diversity of native microbiota that include small contents of LAB, ranging from 2.0 to 4.0 log CFU $g^{-1}$. Therefore, the effective LAB fermentation of fruits and vegetables may require the use of autochthonous or commercially selected starters [45]. The microbial community of fermented vegetables is dominated by LAB genera *Weissella*, *Leuconostoc*, *Pediococcus*, and *Lactococcus* (*Lactiplantibacillus plantarum*,

*Levilactobacillus brevis*, *Leuconostoc mesenteroides*, *Pediococcus pentosaceus*, *Limosilactobacillus fermentum*, and *Lactococcus lactis*) [18,46]. Back-slopping is a traditional, indigenous technique used to ensure the dominance of selected microbiota, contributing to successful plant fermentation (in exp. sauerkraut). Plant-based fermentations can be a suitable environment for the introduction of probiotics, and additionally, the presence of active metabolites including prebiotics (in ex. dietary fibers) can augment the health-beneficial effect. In the field of functional food, vegetables and fruits became an attractive area of development for new products and are also an alternative to dairy products, especially for consumers who are lactose-intolerant, allergic to milk proteins, or vegan [47].

Seasonal vegetables (cabbages, cucumbers, beetroot, carrot, horseradish, olives, radish, onion, garlic, eggplant, zucchini, tomato capers, cauliflower, and young edible bamboo) are traditionally preserved by means of fermentation methods with the involvement of LAB. All plant organs are suitable for fermentation: roots, bulbs, fruit, leaf, stems, and flowers. Among the many fermented plant products, fruit and vegetable juices are becoming increasingly popular. They are obtained traditionally as a result of spontaneous fermentation or by starter culture addition [48]. Fruit and vegetable beverages are preferred by consumers. According to Garcia et al. (2020), the most common fermentation microorganisms of fruit and vegetable juices are species *L. plantarum*, *Lactobacillus bavaricus*, *Lactobacillus xylosus*, *Lactobacillus bifidus*, and *L. brevis.* Fruit and vegetable juices are a promising environment for the delivery of probiotic delivery, therefore ensuring combined health benefits [48]. Nowadays, fermentation of vegetables is widespread at household, artisanal, and industrial levels. Sauerkraut is one of the most common and popular European fermentation products. The fermentation of cabbage is conducted by *Lb. mesenteroides*, *Lb. brevis*, *P. pentosaceus*, and *Lb. plantarum* in anaerobic conditions after the addition of salt. Sauerkraut is usually a product of spontaneous fermentation. Two phases can be distinguished: first, heterofermentative phase with a predominance of *Leuconostoc mesenteroides*, and, second, where the homofermentative *Lactobacillus plantarumin* dominates [18]. In this regard, at least two strains should be applied as a starter culture for the fermentation of high-quality probiotic sauerkraut [49]. Table olives are eagerly eaten and are an ingredient in many dishes. Olives after harvesting due to bitterness are not suitable for consumption. In the process of conversion into table olives, LAB and yeasts are involved. Each olive-growing country follows local processing olive traditions. There are differences in the microbiota of composition of natural black and green olives as well as in lye-treated olives [50].

LAB encountered during olive fermentation are *Lactobacillus*, *Streptococcus*, *Pediococcus*, and *Leuconostoc.* Among *Lactobacillus*, the most frequent are *Lb. plantarum*, *Lb. pentosus*, and *Lb. paraplantarum* [51]. According to the method (in exp. Greek, Spanish, Sicilian, or Californian), LAB and other indigenous microbiota, especially yeast, in various proportions are involved.

The nutritional benefits of plant-fermented products with the contribution of a variety of LAB species potential health benefits respond to the consumers' preferences and food market demands. This is also in agreement with the growing interest of research development. Plant-fermented products, including fruit and vegetable juices, are suitable candidates to be an important food carrier for probiotics [47].

### 3.3. Bakery Products

Since ancient times, sourdough has been utilized as a leavening agent in baking production. This is accomplished by making a sourdough starter from a combination of wheat and water that has been naturally fermented by yeasts and lactic acid bacteria.

Two types of fermentation are used in bakery production: alcoholic, yeast, and lactic—carried out by lactic acid bacteria. Lactic acid fermentation plays a significant role in the bakery industry, particularly in the production of certain types of bread and other baked goods. This fermentation process primarily involves the conversion of sugars into lactic acid by lactic acid bacteria, contributing to changes in homo- or heterofermentation, and offers many technological and nutritional benefits, as well as affecting the quality of bread [52–54].

The genus *Lactobacillus*, which has recently been divided into 23 new genera, is one of the more than 70 bacterial species that have been isolated from sourdough fermentation [17,55].

A traditional method for fermenting cereals used to make bread or steamed bread (mainly in China) is called sourdough fermentation. When compared to the instant active dry yeast frequently used in the food sectors, sourdough-produced bread/steamed bread offers distinctive flavor, texture, nutrition, and shelf-life properties as well as reduced allergenicity [56,57].

In industrial bread production, starter cultures are widely used to obtain good-quality bread. Around the world, sourdough has recently been utilized more frequently as customers' desire for savory, natural, nourishing, and healthy dishes has expanded [58].

The introduction of diverse starter cultures is in line with the current consumer trend of returning to traditional food production methods, as it replaces or limits the use of technological baking additives (e.g., enhancers, enzymes, acids, etc.).

Bakery starter cultures have their own selected bacterial strains that are characterized by the ability to quickly acidify the dough environment, synthesis of compounds shaping the aroma of bread, positively affecting the rheological properties of the dough, bread texture as well as the simultaneous ability to quickly master and colonize the acid environment. Differentiation of the composition of starter cultures is also one way to counteract the unification of bread flavors [59].

Some authors reported that the LAB diversity of sourdough has the potential for geographical indication [60,61].

In Figure 1, the impact of using a starter culture on shortening the process of pre-paring bread dough is presented.

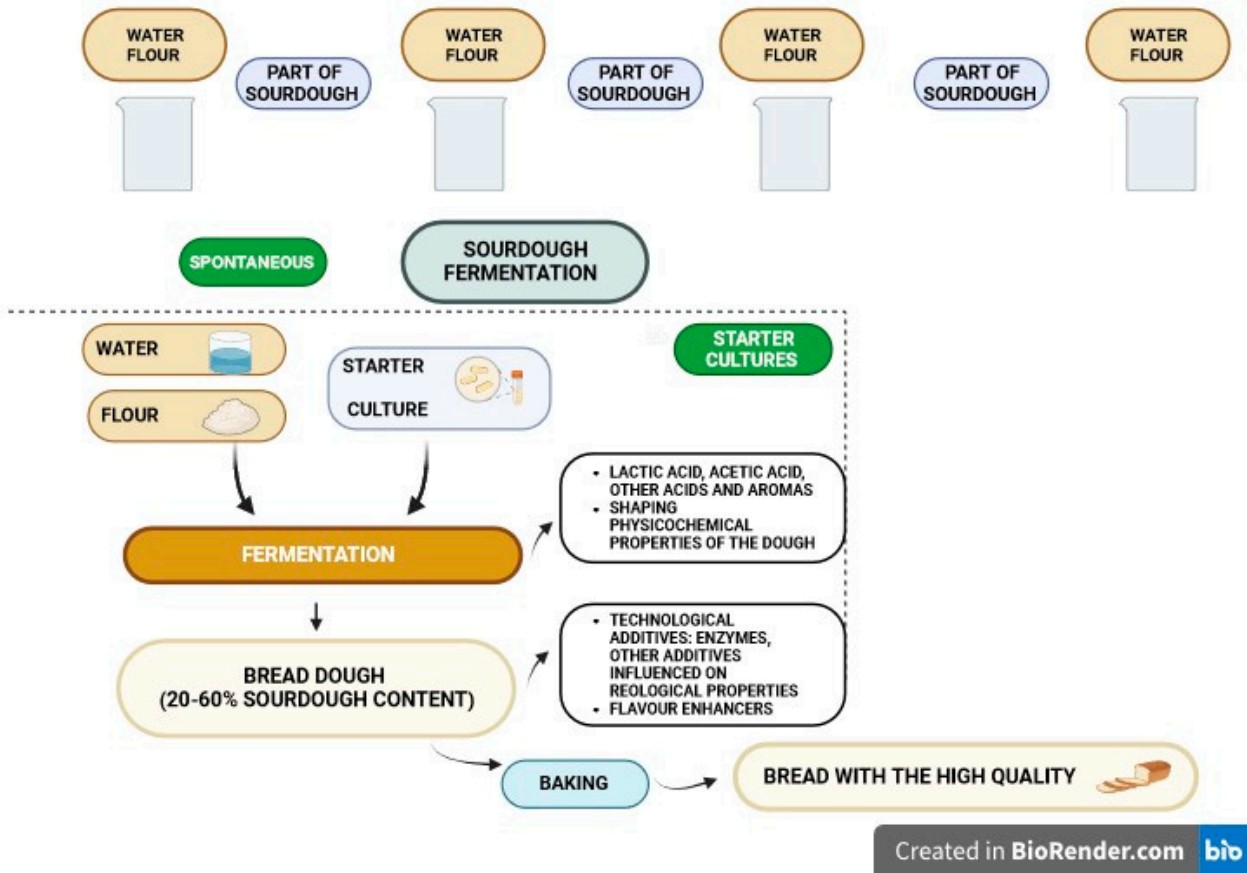

**Figure 1.** Impact of using starter culture on shortening the process of preparing bread dough. Created with BioRender.com.

The use of lactic fermentation in the bakery industry can be considered in many aspects:

(a) Lactic acid fermentation is a key component in the production of sourdough-based bread [62]. In this process, a mixture of flour and water is left to ferment, allowing wild yeast and LAB, naturally present in the environment, to thrive. These microorganisms metabolize the carbohydrates in the dough, producing lactic acid and acetic acid. This acidification of the dough imparts a distinct tangy flavor to sourdough bread and contributes to its unique texture and extended shelf life.

(b) pH Control: Lactic acid fermentation is utilized to control the pH levels in bakery products. By adding lactic acid or using starter cultures with specific LAB strains, bakeries can adjust the acidity of the dough. This helps to improve the texture, flavor, and overall quality of the final product.

(c) Staling Prevention: Lactic acid can also help in preventing staling (the process where bread becomes dry and less palatable) in bakery products. The staling process, in general, explains the mechanism of bread aging and begins immediately after baking [63,64]. When the thermal energy input is stopped, phase transition processes occur, changing the texture of the bread. The recrystallization of amylose within the first few hours after baking has a favorable impact on the solidification of the crumb structure, whereas amylopectin, the second principal macromolecule accounting for the starch percentage in wheat, crystallizes over a longer period of days.

(d) Flavor improving: Lactic acid produced during fermentation contributes to the flavor profile of bakery products. Flavor composition in fermented wheat flour foods depends on some factors, such as fermentation process, cooking procedure, fat oxidation, and also where fermentation by sourdough-associated microbiota plays an important role in geographical indication of cereal [65,66] goods. It imparts a mild tangy taste, which can be desirable in various bread varieties, such as bagels, pretzels, and some types of rolls.

(e) Improving food safety: Lactic acid, along with acetic acid produced during fermentation, has antimicrobial properties. It helps in preserving the freshness of baked goods and inhibiting the growth of harmful microorganisms, extending the shelf life of products [67].

One of the most prevalent dangerous byproducts found in a variety of cooked and fried carbohydrate-rich foods, including baked breads, is acrylamide [68]. Some authors reported that the application of some LAB strains (*Levl. brevis*, *Lacp. plantarum*, *P. pentosaceus*, and *Pediococcus acidilactici*) in baked bread showed significantly reduced acrylamide content [69,70].

Recently, strains synthesizing anti-mold compounds are also sought, which allows one to extend the shelf life of bread [71,72].

During fermentation, LAB causes the raw materials to quickly become acidic, resulting in the production of organic acids, $CO_2$, $H_2O_2$, fatty acids, antifungal peptides, volatile chemicals, and other antifungal substances that prevent the growth of fungi [68]. Furthermore, LAB can use one or more pathways to assist the detoxification of mycotoxins, for example, the use of enzymes and metabolites made by LAB strains, the adsorption of mycotoxins by LAB, or the competition between LAB and other fungi that create mycotoxin [73]. Mycotoxins can be biodegraded by LAB into less poisonous and hazardous chemicals by producing a variety of proteolytic enzymes, such as cell-wall proteinases, peptide transporters, and abundant intracellular peptidases [74]. Additionally, LAB produces citric acid and other organic acids that are highly effective in breaking down aflatoxins.

Overall, lactic acid fermentation is an essential and versatile process in the bakery industry, contributing to the taste, texture, and preservation of various bread and baked goods. It is particularly valuable in the production of sourdough-based bread, where it is responsible for the characteristic flavor and texture.

The principal pathway for the creation of volatile compounds in SD and breadcrumbs is through fermentation, which mostly produces acids, alcohols, aldehydes, esters, and ketones [75]. Lactic acid (sharp acidity) and lactic acid (fresh acidity) are the two main flavors that LAB contributes to in SD bread. This also plays a role in the accumulation of amino acids (such as glutamate, which gives food its umami flavor), as well as the

production of 2-acetyl-1-pyrroline, an end metabolite that forms when ornithine undergoes the Maillard reaction in the arginine deiminase (ADI) pathway and gives food its aroma. Furthermore, the kokumi taste is caused by the accumulation of peptides, such as glutathione and glutamyl dipeptides, which are impacted by LAB activity [76]. Additional flavor molecules can be formed via the conversion of amino acids, such as phenylalanine (sweet), isoleucine (acidic), glycine, serine, and alanine (vinegar/sour) to aldehydes and ketones [75].

### 3.4. Meat and Fish Products

Meat is a very valuable source of food, containing proteins, essential amino acids, minerals, and vitamins (group B). Due to its high nutrition value, moisture content, neutral pH, and possible contamination with pathogens, raw meat and meat products are prone to spoilage. Meat preservation, since ancient times, has been the main concern in human nutrition. Natural preservation methods, including salting fermentation, drying, marinating, smoking, or molding, were the only possible methods ensuring meat storage for a long time in ancient time. Nowadays, chemical preservation additives are applied at industry levels, with possible consumer health disadvantages (in exp. possible carcinogenicity and toxicity of nitrates) and lessening the sensory value of meat products [77]. Fermentation is a traditional, safe, and convenient method of meat bioprotection and conservation, with no negative effects on human health. The most common meat biofermentating agent is LAB, which is usually present in raw meat [78]. LAB metabolites have antibacterial and antioxidant activity; moreover, they contribute to nitrate degradation. During fermentation, LAB produces lactic acid, and the meat acidity increases, which, along with LAB bacteriocins and organic acids, hydrogen peroxide, and diacetyl, inhibits the pathogenic and spoilage microorganism occurrence [10]. Historically, the main purpose of meat biopreservation was meat storage; however, nowadays, for consumers, the key aspects are unique organoleptic properties of fermented meat products, including unique taste, aroma, texture, and color [79].

Traditional meat products subjected to spontaneous fermentation by unspecified autochthonous microbiota have unique local sensory characteristics. The native microbiota is well adapted to meat matrices and intrinsic and extrinsic factors occurring during fermentation and maturation [80]. For hundreds of years, a variety of regional, artisanal fermented meat products have been created [81]. The manufacturing of fermented sausages requires the addition of salt and nitrates; therefore, the fermentative microorganism should be adapted to such an environment. Subsequent drying, smoking, and temperature modifications depend on the used formula. Manufacturing conditions favor the growth of LAB, like *Lactobacillus sakei*, *Lactobacillus curvatus*, *Pediococcus pentosaceus*, and coagulase-negative staphylococci, like *Staphylococcus xylosus*, *Staphylococcus equorum*, *Staphylococcus warneri*, and *Staphylococcus succinus* [82]. However, there are regional and geographical differences in meat-manufacturing processes, including salt levels, temperature, pH as well as bacterial diversity. In a study of 80 randomly selected European fermented meat products, Van Reckem et al. (2019) showed that, in Northern Europe, fermented meat had lower pH (even less than pH 5), was fermented at higher temperatures, and salt levels varied from 2% to 6%. The prevalence of *Pediococcus pentosaceus* and *Staphylococcus carnosus* in Northern European fermented meat products is higher, while in less-acidified fermented meat of Southern European origin, more frequently *Lactobacillus sakei*, *Staphylococcus xylosus*, and *Staphylococcus equorum* are present [83].

Commercialization of fermented meat products reflects growing consumer demands. For industrial fermented sausage production, starter cultures are used to achieve safety and uniformity of the final product. Selected microorganisms of starter cultures should tolerate the presence of acid, sodium nitrite, and sodium chloride, and must be able to dominate the product's microflora. In this regard, the preferred source for isolation of bacteria for starter cultures is traditional fermented meat [84]. Starter cultures designed for meat fermentations consist of LAB (in exp. *Lb. sakei*) to ensure proper acidification safety as well as flavor, taste,

color, and texture, and other bacteria strains (*Staphylococcus*, *Pediococcus*, *Kocuria*), yeasts, or molds to provide unique organoleptic properties [2,85]. Recently, there has been growing interest in the introduction of probiotics into meat products. Reports of the successful introduction of various probiotics strains (*Lactobacillus plantarum*, *Lactobacillus paracasei*, and *Lactobacillus rhamnosus*) have shown that fermented meat is a probiotic food of great potential [86].

Fish fermentation is the traditional technique of fish preservation used by people in coastal regions. Fish is a valuable source of protein and fatty acids, including essential omega-3 fatty acids, minerals, and vitamins. Fermented fish and fish products have unique sensorial proprieties appreciated by consumers. According to the FAO report in 2019, aquatic foods provided about 17 percent of animal proteins worldwide [87]. Due to the action of autolytic enzymes and microbe activity, the spoilage of fish meat is usually rapid. Therefore, the preservation of fish meat is essential. It consists of several stages. Salting is very important. The hypersaline environment deteriorates enzymatic activity and facilitates the growth of properly fermented microorganisms.

Different organoleptic characteristics depend on the type of fish fermentation method: spontaneous with natural microbes or with the use of different starter cultures, and as well as to specific environmental conditions, and ingredients or spices added. Fish can be fermented whole, sliced, or in the form of paste [88]. Salt has a pivotal role in the preservation of fish meat. It reduces water activity and the development of pathogenic and spoilage microorganisms. According to the amount of added-salt, high-salt products (salt content >20%) and low-salt products (salt content: 3–8%) can be distinguished [89]. In low-salt fermented fish, LAB is predominant. In high-salt products, halophilic lactic acid bacterium *Tetragenococcus* spp. resistant to salt stress is predominant [90].

The use of selected starter cultures in the fish industry is limited. Most fermented fish products are from spontaneous fermentation. According to the analysis of Belleggia and Osimani (2023) of data from 168 research papers, lactic acid bacteria are predominant in most fermented fish and fish-based products, with counts up to 9.5 log CFU g$^{-1}$ [91]. Among LAB, the most important fermentative strains are *Lactiplantibacillus plantarum*, *Latilactobacil-lus. sakei*, and *Latilactobacillus curvatus*. Other predominant microorganisms present in fermented fish are *Staphylococci*, *Bacillus* spp., and yeasts [91].

*3.5. Oriented Fermented Food Products*

Fermentation is an ancient technique for food preservation and transformation and also has a significant role in traditional culinary cultures [92]. The historical significance of oriental fermented food products dates back centuries, and their origins are deeply rooted across Asia, including China, Japan, Korea, and Southeast Asian nations. Oriental fermented food products are valuable with their unique flavors, potential health benefits, and extended shelf life. Producing these products requires traditional knowledge, microbiology, and culinary expertise. These food products include a variety of raw materials, such as milk, cereals, millet, vegetables and meat [93–95]. The fermentation techniques include balance of ingredients, temperature, and time control, which can be different from one region to another and result in regional specialties [96].

Oriental fermented foods often rely on naturally occurring microorganisms, including bacteria, yeasts, and molds [97]. That is why it is common for oriental cultures to ferment their foods in specific types of fermenting vessels, such as earthenware pots or wooden barrels, to ensure a stable microbiological environment for fermentation [98]. Lactic acid fermentation is the most essential way to prepare traditional fermented oriental foods, contributing to their unique flavor profiles, textures, and nutritional compositions [93].

Tarhana is a fermented cereal-based soup mix and includes a combination of grains, vegetables, and yoghurt from Turkey [99]. Fermentation methods for tarhana can be divided into two types: lactic acid fermentation with yoghurt addition and lactic acid fermentation combined with yeast fermentation (baker's yeast addition). Lactic acid fermentation determines the sensory characteristics of tarhana; it gives it a sour and acidic flavor [100].

Lactic acid bacteria have been identified in different tarhana samples: *Pedicoccus acidilactici*, *Streptococcus thermophilus*, *Lactobacillus fermentum*, *Entericoccus faecium*, *Leuconostoc pseudo-mesenteroides*, *Weissella cibaria*, *Lactobacillus plantarum*, *Lactobacillus bulgaricus*, *Leuconostoc citreum*, *Lactobacillus paraplantarum*, *Lactobacillus casei*, *Lactobacillus brevis*, and *Lactobacillus plantarum* [101].

Another fermented product is Boza, which is made from grains, like millet, wheat, or corn, and is a popular fermented cereal-based beverage in Turkey and the Middle East. During the fermentation process, starches are broken down into sugars and lactic acid, giving a sour and sweet taste to the drink [102]. Boza is often enjoyed during special occasions and festivals. The identified strains during fermentation are *Lactobacillus paracasei*, *Lactobacillus plantarum*, and *Lactococcus lactis* [103]. In another study, various kinds of lactic acid bacteria strains were isolated from Boza samples, including newly identified species, such as *Lactococcus garvieae*, *Pediococcus parvulus*, and *Streptococcus macedonicus* [104].

Stinky tofu is also another traditional fermented food in Asian cuisine. The fermentation of tofu is made in a brine solution, and it gives a distinct odor and flavor. Stinky tofu is typically deep-fried or grilled for various dishes [105]. According to studies for the identification of the microbial composition of stinky tofu, the most dominant strains are belonged to *Lactobacillus* and *Bacillus* genera [106]. In addition, Chao et al. conducted an analysis and identified a total of seven different genera and thirty-two species, including *Enterococcus*, *Lactobacillus*, *Lactococcus*, *Leuconostoc*, *Pediococcus*, *Streptococcus*, and *Weissella* [107]. These findings underline the diverse microbial community present in stinky tofu and its fermentation process. The texture of stinky tofu can differ from soft to firm for possibilities in culinary experimentation.

Natto is a traditional Japanese dish that is made through the fermentation of soybeans with *Bacillus subtilis* [108]. The fermentation of natto produces several bioactive components, including nattokinase, daidzein, phytosterols, superoxide dismutase, and several biologically active peptides, improving nutritional quality [109]. Miso is a seasoning made from fermenting soybeans and grains with koji mold (*Aspergillus oryzae*) and LAB strains. After yeast fermentation, LAB leads to a sour and umami taste with increasing acidity. Miso can be of several types, such as white, red, and mixed, each one of them offering different flavors suited for soups, marinades, and dressings [110]. In recent years, soy sauce has become an increasingly popular oriental fermented condiment. Lactic acid bacteria with other microorganisms such as molds and yeasts break down the complex proteins and carbohydrates in soybean into simpler ones. This fermentation process enhances the umami flavor of soy sauce. Moreover, the presence of LAB in these products offers potential probiotic benefits [110,111].

Fermentation preserves vegetables as well as enhancing their flavor, texture, and nutritional value [112]. Fermentation can naturally occur by their own lactic acid bacteria, such as *Lactobacillus*, *Leuconostoc*, and *Pediococcus* [48].

For example, kimchi is a side dish from fermented vegetables, such as napa cabbage and radishes in Korean cuisine. This side dish is rich in fiber, vitamin A, riboflavin, vitamin C, thiamine, potassium, iron, and calcium, also with a low-calorie count. In addition, the nutritional value of raw material kimchi produced with spontaneous fermentation includes different kinds of lactic acid bacteria [113]. *Lactobacillus*, *Weissella*, *Lactococcus*, *Pediococcus*, and *Leuconostoc* were isolated from kimchi samples [114]. Due to its health benefits and nutritional value, kimchi is considered in the list of the five healthiest foods in the world. The other four are Indian lentils, Spanish olive oil, Greek yoghurt, and Japanese soybeans [115]. Regular consumption may contribute to heart health, regulating cholesterol levels and blood pressure. These factors make kimchi not only a flavorful addition to meals but also a valuable component of a health-conscious diet [116].

Similarly, sauerkraut, meaning "sour cabbage", is a traditional fermented vegetable dish known for its distinctive taste and long shelf life. The production process involves shredding fresh cabbage and mixing it with 2.3–3.0% salt, initiating natural fermentation. In the beginning of fermentation, *Leuconostoc* spp. starts the process and then *Lactobacillus*

spp. and *Pediococcus* spp. complete the process. The final product has a pH of 3.5 to 3.8. Fermentation gives sauerkraut its unique taste, as well as preserving and adding nutritional value [112].

Chinese-style sausage is a popular indigenous fermented meat product in China. Lactic acid bacteria play a key role in enhancing the flavor of the meat. The fermentation process produces lactic acid by decomposing carbohydrates, leading to a decrease in pH within the meat [117]. This acidic environment is unsuitable for the growth of food-borne pathogens, making it an effective natural preservative (Wang, 2019) [118]. Nasezushi is a traditional Japanese fermented meat product that is made with salted fish and cooked rice and ferments spontaneously for more than a month [119]. Studies have identified the presence of beneficial bacteria, like *Lactobacillus acidipiscis*, *Lactobacillus versmoldensis*, *Lactobacillus plantarum*, *Tetragenococcus muriaticus*, and *Tetragenococcus halophilus*, in narezushi [120,121]. Furthermore, narezushi has been found to have inhibitory effects on *Listeria monocytogenes* infection, highlighting its potential as a food with natural antimicrobial properties [122]. Another notable Chinese fermented fish product is Chouguiya, also known as Stinky Mandarin fish. This fish-based fermented food contains a variety of bacteria, including *Lactobacillus sakei*, *Lactococcus garvieae*, *Lactococcus lactis*, *Lactococcus raffinolactis*, *Vagococcus* species, *Enterococcus hermanniensis*, *Macrococcus caseolyticus*, and *Streptococcus parauberis*. The unique fermentation process gives Chouguiya its distinct flavor and character, making it a significant part of Chinese culinary heritage [123].

The microbiological complexity and potential health benefits of oriental fermented foods have attracted scientists. Several studies have revealed the diverse range of microorganisms involved in fermentation, providing insight into the microbial composition of these products. Further, investigations into the bioactive compounds produced by fermentation can have an impact on potential applications in functional foods and nutraceuticals.

## 4. Probiotics as Novel Microorganisms in Lactic Acid Fermentation of Food

The growing consumer interest in food that is "natural", "without preservatives" and, at the same time, "high quality" makes food producers look for new technological solutions to meet these market requirements. Today, however, bioconservation is defined as using natural or isolated primary and/or secondary metabolites from sources, such as bacteria, fungi, plants, and animals, for food preservation. Examples of bioconservants are oils and plant extracts or enzymes of plant and animal origin [124].

The term "probiotic microorganisms" is defined as "live microorganisms that, when administered in adequate amounts, confer a health benefit on the host" [125]. Microorganisms should be precisely defined in order to be classified as probiotic strains by defining appropriate criteria for safety of use, functional and technological characteristics. Microorganisms—candidates for "probiotic"—must meet three key requirements: (1) must be alive at the time of administration and must be microorganisms, (2) must be administered in a sufficiently high dose to have a health-promoting effect, and (3) must have a beneficial effect on the host [126,127].

Our intestinal microbiota includes lactic acid bacterial strains, which are also frequently utilized as probiotics. In addition to *Lactobacillus* and *Bifidobacterium*, other significant and prevalent probiotic microorganisms include *Lactococcus*, *Streptococcus*, *Enterococcus*, *Propionibacterium*, and *Saccharomyces* yeasts [128,129].

The mechanisms of action of probiotic bacteria, thanks to which they have a positive effect on the human body, have not yet been fully understood. However, some scientific reports indicate, in this case, such factors as: competitiveness in relation to intestinal pathogens, neutralization of anti-nutritional and carcinogenic factors, production of metabolites with antibacterial activity and modulation of the mucosa and systemic immune system [126,127,130].

Many health benefits of LAB bacteria include alternatives to antibiotic, lowering cholesterol level, inhibition of harmful microorganisms in the gut, and boosting the immune

system, making them promising probiotic candidates, which are studied to explore their potential applications and affect the safety in fermented food products [131].

The diversity of microorganisms found in naturally fermented foods makes them an excellent source of potential probiotics [132].

Until recently, strains of probiotic bacteria were isolated only from the gastrointestinal tract of healthy people. In recent years, interest in new strains isolated from other unconventional sources has been growing, which are also able to survive the difficult conditions in the human digestive tract [133]. Some researchers concluded that probiotic microbes can also come from unusual sources, like animal digestive tracts, breast milk, food (both fermented and unfermented), air, or soil, in addition to the traditional source (a healthy human digestive system). In particular, for the advancement of technology for the creation of food-specific vaccines, the isolation, identification, and assessment of safety and probiotic qualities of novel, "wild" strains of microbes from traditional food encompass a vital technique. In addition to their defensive (bactericidal and bacteriostatic) qualities, new vaccinations may also have benefits for enhancing consumer health [134–137]. It seems acceptable to expand the definition of "probiotic" to include microorganisms isolated from conventional, naturally fermented food. The microflora of the environment in which the products were produced is made up of microorganisms that have been isolated from fermented products. They could make an intriguing alternative to gut bacteria if studied, especially in terms of their probiotic capabilities and safety. The production of fermented foods today is based on the use of lactic acid bacteria as starter cultures in order to initiate and provide controlled and predictable [138] fermentation (Figure 2).

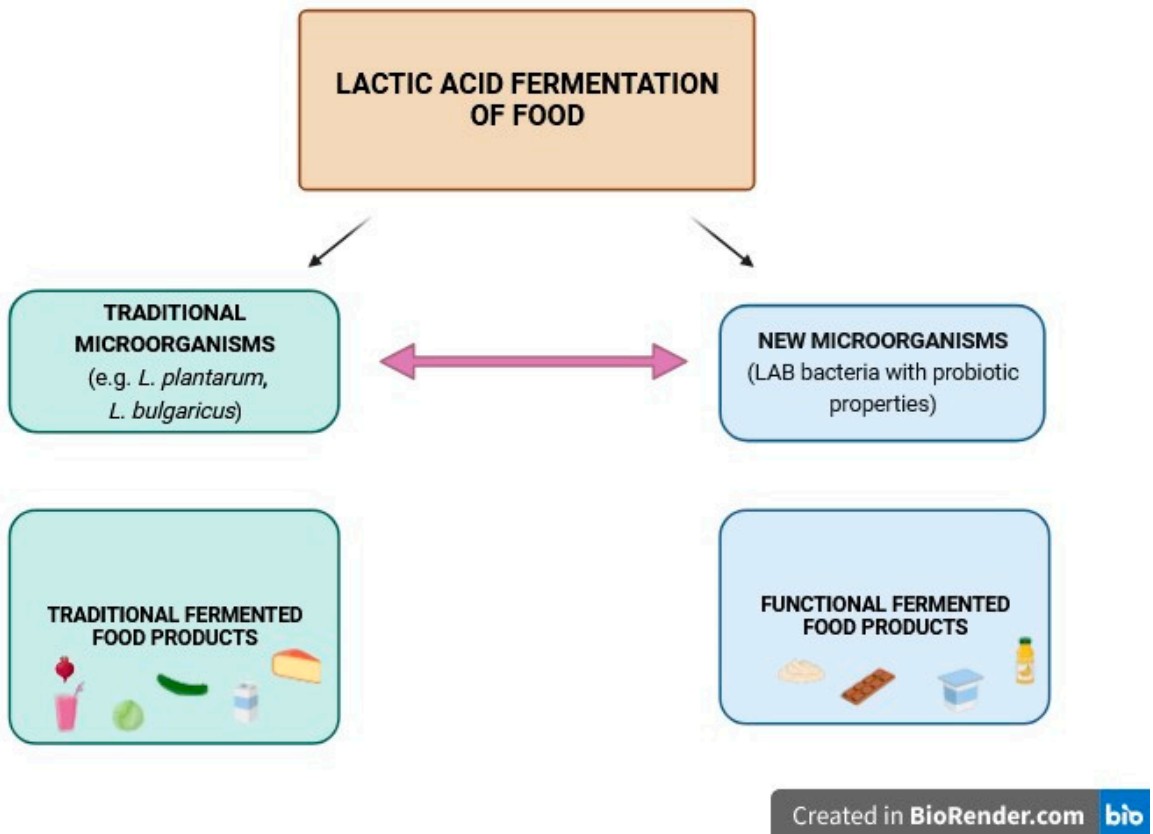

**Figure 2.** Lactic acid microorganisms used in fermented foods—overview. Created with BioRender.com.

When compared to spontaneous fermentation, the use of starter cultures can provide a number of advantages, including improved overall fermentation control, a shorter ripening period, a lower risk of the growth of pathogenic microorganisms, and improved quality of preservation between batches [80,139].

Consumer interest in the probiotic food segment has set a new trend related to sales by specialized companies of the so-called "starter cultures" for food production. Thanks to the probiotic properties of the microflora used, this industry is developing towards health-promoting food. It is also important for numerous research laboratories around the world to work on designing new technological solutions in the field of fermented food production, with the participation of microorganisms with probiotic properties, which enhances consumer confidence in this type of food in the daily diet [140–143].

It is important to note that not all probiotics are suitable for lactic acid fermentation, and the choice of probiotic strains should be carefully considered. Bacterial strains with probiotic properties, used in lactic acid fermentation, should be selected based on their ability to thrive and ferment the specific food matrix and their compatibility with the desired flavor profile.

Food matrices are raw materials, into which probiotic bacteria can be introduced to enable their development and acceptance of the food product. Introduced probiotic bacteria may cause fermentation or remain alive in the unfermented product.

Characteristics of individual probiotic strains, food matrices, and dietary interactions between them are new areas of food research for technologists. On the other hand, factors affecting the viability of probiotic bacteria in probiotic products include product parameters, such as pH, titration acidity, oxygen, water activity, the presence of salt, sugar, and other compounds (peroxide hydrogen, bacteriocins, artificial flavorings and coloring agents, etc.), processing parameters, including fermentation conditions (incubation temperature, heat treatment, product cooling and storage conditions, packaging materials, production scale), or, finally, microbiological parameters (strain of probiotic used, rate and proportion of implantation) [144,145]. The main problems associated with designing probiotic food are as follows:

- Providing conditions in the food matrix that will guarantee the viability of probiotics, i.e., their growth and/or survival during food processing and storage, and at the same time maintaining a beneficial effect on health;
- Ensuring adequate sensory properties of the product;
- Ensuring a sufficiently high number of probiotic bacteria. The probiotic features and health benefits conferred are known to be LAB-strain-specific [146]. Probiotic food products must have a high concentration of microorganism cells ($\geq 10^6$ colony-forming units (CFU mL$^{-1}$), or between $10^8$ and $10^{11}$ colony-forming units (CFU) per day), to have the required positive impact [147]. Such a high number of bacterial cells should persist throughout the shelf life of food. Probiotic cells must survive the passage through the gastrointestinal tract, reach the colon in sufficient numbers, and, finally, adhere to, and colonize the gut epithelium [148–150].

Products containing probiotics must undergo a number of technological processes that will properly prepare them for action in the human body after ingestion. At the beginning of the design of the probiotic product, it should be decided to which group of society it will be dedicated. This action will allow for the selection of an appropriate probiotic strain that has proven action based on scientific evidence in a given disease entity [151].

Another important aspect of the choice of probiotic bacteria is their characteristics and functional screening. This step is necessary because it allows for the examination of the individual characteristics of a given species of probiotic bacteria and assesses which of them meets the criteria assumed by the manufacturer. The screening models used in this step include both simple in vitro cell tests and more complex ex vivo tests or animal model tests. Another important stage in the selection of microorganisms for the production of probiotic products is the identification and assessment of the safety of probiotic strains.

After choosing the strain and examining its safety, another important stage is an assessment of whether the production process of a given strain will be possible in the right quantity. This is an important stage in production because, in end products, probiotics must be stable and have consistent performance with the intended use [151]. This is crucial for high-quality probiotic preparations with doses established based on clinical trials, with

a large number of cells and a long duration in many different temperature and humidity conditions. By controlling the production process and, thus, the quality of each strain in the final probiotic preparation, it is common practice that the cell mass of each strain is produced separately [151].

Fermented food producers place high demands on starter cultures used in food processing, which increases the possibility of adapting microorganisms to more effective food fermentation. Progress in biotechnology and genetic engineering allows for the discovery of new strains of probiotic microorganisms, which has an innovative impact on the basic technological processes used, i.e., lactic acid fermentation, and the obtained products are characterized by broadly understood functionality [152–155].

## 5. Conclusions

In summary, this article provides in-depth knowledge about the evaluation of the lactic fermentation process over the years, depending on the nature of the microflora used. Lactic acid bacteria (LAB) are commensal habitants of the human gastrointestinal tract (GIT). They have a long history of being used in foods and fermented products as starter cultures.

Nowadays, the increasing use of LAB bacterial strains isolated from different sources as starters has attracted the attention of researchers. Carrying out the lactic fermentation process with the participation of these new microorganisms with probiotic properties, supporting the reconstruction of the quantitative and qualitative balance of microorganisms inhabiting the intestines during dysbiosis, enables linking the issue of food preservation, with increasing health value in the final product.

Probiotics can serve as novel microorganisms in the lactic acid fermentation of food, offering the dual benefits of improved preservation and enhanced nutritional value, especially in terms of promoting gut health. The choice of probiotic strains and their application in various food products depends on the specific desired outcomes and consumer preferences.

To conclude, using starter cultures in the course of the lactic fermentation process seems to be a promising alternative to improving the technological and functional properties of final food products.

**Author Contributions:** Conceptualization, D.K.-K., B.S. and A.S.; methodology, A.S., B.S. and K.K.; software, A.S., B.S. and K.K.; validation, A.S. and B.S.; formal analysis, D.K.-K., B.S. and A.S.; investigation, B.S., A.S. and K.K.; resources, A.S. and B.S.; data curation, A.S. and B.S.; writing—original draft preparation, B.S., A.S., K.K. and D.K.-K.; writing—review and editing: B.S., A.S. and D.K.-K.; visualization, B.S. and A.S.; supervision, B.S. and A.S. All authors have read and agreed to the published version of the manuscript.

**Funding:** This research received no external funding.

**Institutional Review Board Statement:** Not applicable.

**Informed Consent Statement:** Not applicable.

**Data Availability Statement:** Not applicable.

**Conflicts of Interest:** The authors declare no conflict of interest.

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
