# Peer review of "Traditional and New Microorganisms in Lactic Acid Fermentation of Food"

_fermentation, doi:10.3390/fermentation9121019_

Round 1

Reviewer 1 Report

Comments and Suggestions for Authors

The manuscript entitled “Traditional and new microorganisms in lactic acid fermentation of food”summarized the recent research progress in traditional fermentation technology and fermented food. 

Fementaiotn is an ancient and traditional way to preserve and/or improve the flavor of food, and lactic acid bacteria are most abundant microorganisms discovered in fermented food. 

This manuscript made a good summary about traditional fermentation related to lactic acid fermentation.

One of the concern is that most of the references are from European countries, authors please add some more information about researches done in traditional fermented food in Asian countries.

Authors better provide a graph describing the overview of lactic acid microorganisms found or used in fermented foods.

Comments on the Quality of English Language

The language should be further improved. Current version is a little difficult to understand. 

Author Response

Response to Reviewer 1:

We would like to thank the Reviewer for careful and thorough reading of this manuscript and for  the thoughtful comments and constructive suggestions. Our responses to each comment have been marked. All the changes in manuscript were marked in colors.

The manuscript entitled “Traditional and new microorganisms in lactic acid fermentation of food” summarized the recent research progress in traditional fermentation technology and fermented food.

Fermentation is an ancient and traditional way to preserve and/or improve the flavor of food, and lactic acid bacteria are most abundant microorganisms discovered in fermented food.

This manuscript made a good summary about traditional fermentation related to lactic acid fermentation.

Comment 1: One of the concern is that most of the references are from European countries, authors please add some more information about researches done in traditional fermented food in Asian countries.

Response: We thank you for this suggestion. The subsection no. 3.5 has been enriched with additional information on Asian fermented foods (lines: 604-627). The suitable references are in lines: 1063 -1081.

Comment 2: Authors better provide a graph describing the overview of lactic acid microorganisms found or used in fermented foods.

Response: thank you for this suggestion. The correction has been made. In the text of the manuscript, Figure 2 was added (lines: 680-682).

Comment 3: The language should be further improved. Current version is a little difficult to understand.

Response: the language of presented manuscript was checked and improved.

Reviewer 2 Report

Comments and Suggestions for Authors

The manuscript is very interesting and comprehensive. It is worth to be published. But I noticed one big mistake: it seems to me that most of the references in the main text are wrong written considering the number belonging. You missed reference number 28, and afterward, each reference is one number lower when you compare it with the list of references. You can see it easily when you find in the text authors Garcia et al. (2020) who are presented by reference number 48, but in the list of references, they are under the number 49 (page 6, line 289). Please change Lb. into Lc. for Lc. lactis subsp. cremoris (page 5, line 234). Some words in the text are broken by a dash. I identified many in the file in the attachment (please see). Instead of the word business, I suggest production (page 7, line 320). The resolution of  Figure 1. is poor. In the text, Rekem needs to be changed to Van Reckem (page 10, line 458). The manuscript needs some minor revision by the authors before publishing. 

Author Response

Response to Reviewer 2:

We would like to thank the Reviewer for careful and thorough reading of this manuscript and for  the thoughtful comments and constructive suggestions. Our responses to each comment have been marked. All the changes in manuscript were marked in colors.

The manuscript is very interesting and comprehensive. It is worth to be published.

Comment 1: But I noticed one big mistake: it seems to me that most of the references in the main text are wrong written considering the number belonging. You missed reference number 28, and afterward, each reference is one number lower when you compare it with the list of references. You can see it easily when you find in the text authors Garcia et al. (2020) who are presented by reference number 48, but in the list of references, they are under the number 49 (page 6, line 289).

Response: We thank you for this suggestion.  

At item 16, the text  "(Accessed on 10 October 2023)" was additionally marked  as number 17 by mistake. This disturbed the whole numbering.  The reference 28 was added in line 194.

The numbering of literature items has been changed according to the reviewer’ suggestion.

Comment 2: Please change Lb. into Lc. for Lc. lactis subsp. cremoris (page 5, line 234).

Response: the correction has been made (lines:  245-246).

Comment 3: Some words in the text are broken by a dash. I identified many in the file in the attachment (please see).

Response: We thank you for this suggestion. The whole text of manuscript was checked and improved taking into account this aspect.

Comment 4: Instead of the word business, I suggest production (page 7, line 320).

Response: the correction has been made (line 333).

Comment 5: The resolution of  Figure 1. is poor.

Response: We thank you for this suggestion. The page 9 shows Figure 1 in the new resolution.

Comment 6: In the text, Rekem needs to be changed to Van Reckem (page 10, line 458).

Response: the correction has been made (line 478).

The manuscript needs some minor revision by the authors before publishing.

Reviewer 3 Report

Comments and Suggestions for Authors

  • The submitted manuscript has no novelty. It seems only gathering information from the cited paper without any purpose. I went briefly to check several papers about similar topic, I found the papers below are almost similar I would say.

  • https://doi.org/10.1080/10408398.2021.2025035

  • https://doi.org/10.1080/10408398.2021.1929059

  • https://doi.org/10.1080/10408398.2020.1858269

  • https://doi.org/10.1111/1541-4337.12897

  • https://doi.org/10.1016/j.cofs.2021.11.013

  • https://link.springer.com/article/10.1007/s43393-021-00044-w

  •  
  • However, those papers are not even cited in this submitted manuscript. Therefore, I doubt the update of this manuscript for publication. Therefore, I suggest rejecting this manuscript for publication.

Author Response

Response to Reviewer 3:

We would like to thank the Reviewer for the careful and thorough reading of this manuscript and for the thoughtful comments and constructive suggestions. Our responses to each comment have been marked. All the changes in the manuscript were marked in colors.

Comment 1:   The submitted manuscript has no novelty. It seems only gathering information from the cited paper without any purpose. I went briefly to check several papers about similar topic, I found the papers below are almost similar I would say.

  1. https://doi.org/10.1080/10408398.2021.2025035
  2. https://doi.org/10.1080/10408398.2021.1929059
  3. https://doi.org/10.1080/10408398.2020.1858269
  4. https://doi.org/10.1111/1541-4337.12897
  5. https://doi.org/10.1016/j.cofs.2021.11.013
  6. https://link.springer.com/article/10.1007/s43393-021-00044-w

    However, those papers are not even cited in this submitted manuscript. Therefore, I doubt the update of this manuscript for publication. Therefore, I suggest rejecting this manuscript for publication.

Response: We thank you for your suggestions.

The article is based on a historical outline, from the time when the purpose of fermentation was to extend the shelf life of food products to modern times, in which the basic advantages are taste and health benefits. It shows the role and importance of LAB in fermentation processes. This is a summary with examples of fermented foods from around the world, with a focus on Europe and Asia. Such a broad presentation of fermented food and microorganisms traditionally involved in the process of lactic acid  fermentation, aims to show its cultural and microbiological richness. Individual issues, such as biochemical changes that have a potential impact on the taste and other qualities of food, the importance of non-starter LAB or even issues in the field of genetic engineering have not been described in detail, because that was not the purpose of the work. As rightly noted by the Reviewer, these topics are discussed in other review papers that have been indicated by the Reviewer.

However, the aim of this study was to show, on this  background that in the case of fermented food, the selection of a suitable food matrix and microorganism (e.g. probiotic microorganisms) is crucial. The main ideas and objectives that concern the design and technology of fermented food production are described and analyzed in the final part of the review. An innovative and justified in the content of the article is the fact that fermented food using LAB combines a number of benefits, such as food preservation, traditional taste qualities, and high nutritional value with a beneficial effect on consumer health. Currently, new microorganisms (including probiotic ones) are increasingly used to produce food with health-promoting properties.

This interpretation is also presented in Figure 2. It was added in the revised of the manuscript (lines: 680-682).